# S$^2$C$^2$IL: Self-Supervised Curriculum-based Class Incremental Learning

## Abstract

Class-incremental learning, a sub-field of continual learning, suffers from catastrophic forgetting, a phenomenon where models tend to forget previous tasks while learning new ones. Existing solutions to this problem can be categorized into expansion-based, memory-based, and regularization-based approaches. Most recent advances have focused on the first two categories. On the other hand, limited research has been undertaken for regularization-based methods that offer deployability with computational and memory efficiency. In this paper, we present Self-Supervised Curriculum-based Class Incremental Learning (S$^2$C$^2$IL), a novel regularization-based algorithm that significantly improves class-incremental learning performance without relying on external memory or network expansion. The key to S$^2$C$^2$IL is the use of self-supervised learning to extract rich feature representations from the data available for each task. We introduce a new pretext task that employs stochastic label augmentation instead of traditional image augmentation. To preclude the pretext task-specific knowledge from being transferred to downstream tasks, we leave out the final section of the pre-trained network in feature transfer. In the downstream task, we use a curriculum strategy to periodically vary the standard deviation of the filter fused with the network. We evaluate the proposed S$^2$C$^2$IL using an orthogonal weight modification backbone on four benchmark datasets, split-CIFAR10, split-CIFAR100, split-SVHN, and split-TinyImageNet and two high-resolution datasets, split-STL10, and ImageNet100. The results show that S$^2$C$^2$IL archives state-of-the-art results compared to existing regularization-based and memory-based methods in class-incremental learning algorithms.

## 1 Introduction

The concept of incremental learning has been an active area of research in the deep learning community (Mai et al., 2022; Delange et al., 2021). Humans have an unparalleled capability to incrementally capture information of new tasks, domains, or classes without forgetting the knowledge gained from past episodes. As shown in Figure 1, deep learning approaches, especially CNNs, show excessive plasticity in learning new tasks though they lack the inherent tendency of incremental/continual learning. The problem of incremental learning is introducing new knowledge to an existing model calibrated with old knowledge. During the introduction of new knowledge, one of the biggest challenges is to retain the information from old knowledge as CNNs tend to lose previously acquired information leading to a phenomenon termed as *Catastrophic Forgetting* (McCloskey & Cohen, 1989; Ratcliff, 1990; McClelland et al., 1995; French, 1999). To mitigate catastrophic forgetting, researchers have proposed different approaches to overcome catastrophic forgetting in deep learning models. These approaches include expansion-based models, in which new parameters are added to the network with the addition of each task which capacitates the model to aggregate the information of new classes subsequently. Other approaches include using memory-based methods, where a model is trained to continually relearn old information to retain memory stability with external memory banks, and regularization-based techniques to prevent the model from overwriting previously learned knowledge. Additionally, some researchers have proposed using a combination of these methods, as well as other techniques, in order to more effectively address the problem of catastrophic forgetting and enable deep learning models to continue to learn and adapt over time. Overall, developing effective approaches to overcoming catastrophic forgetting is crucial for advancing deep learning and the continued development of more intelligent and adaptive models.

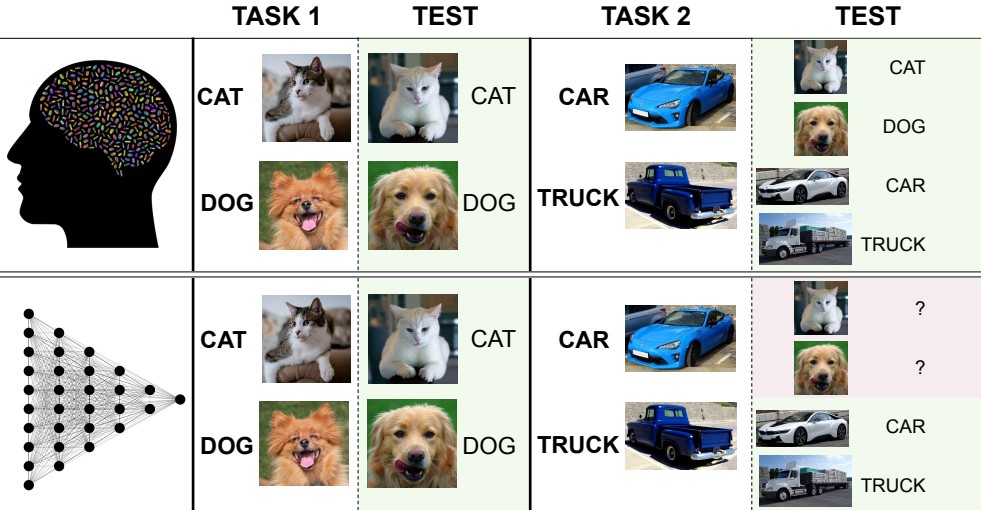

Figure 1: Humans can easily generalize over a newer set of classes/tasks, whereas neural networks suffer from the problem of catastrophic forgetting when trained for different tasks over time.

This work presents a novel regularization-based continual learning algorithm focusing on class-incremental learning (CIL). In general, the performance of these approaches is inferior to memory-based. However, they are more efficient computationally, maintaining learning plasticity without needing additional memory to maintain stability. They are also easy to deploy as opposed to expansion-based and memory-based approaches where voluminous memory setup is a requisite. Without such memory reserves, regularization-based approaches depend mostly on utilizing the data available for the current incremental task. Continual learning models can only extract features necessary from limited data for their current task, causing a loss of prior information and inability to perform joint classification with previously learned tasks. This phenomenon is termed as *Allocentric Ignorance*. One of the ways to alleviate this issue is through unsupervised learning. Unsupervised pre-training aids the model in learning distinct features from limited or unlabelled data, which assists the model in better generalization to newer incremental tasks. To this end, we propose a novel Self-Supervised Learning (SSL) task termed as *Stochastic Label Augmentation (SLA)*, which optimizes the model to extract a diverse set of features from the limited data for the current task. The task synthesizes information from different parts of an image using stochastically generated labels in a multitask fashion, thereby providing rich and diverse feature representations to the model. Conventionally, the data is augmented in self-supervised learning to generate proxy labels that protract the training and are computationally inefficient. By fitting the pretext dataset on stochastically generated labels, we improve the computational efficiency of the model. We employ a novel task-wise weight-regularizer in the pretext task that prevents information loss from the previous to the next incremental task and keeps the model bound to the previously acquired knowledge. Further, we mitigate the transfer of pretext task-specific knowledge to the downstream task through *Penultimate Weight Sharing (PWS)* between networks.

After pre-training the network for the classification task in the CIL setting, we employ a *curriculum-based learning* technique during the downstream training. Curriculum Learning-based algorithms facilitate faster convergence and enhanced generalizability by ordered training (Bengio et al., 2009). A Gaussian kernel is used to smooth the extracted features, with the smoothing effect increasing periodically. With the reduction in information through smoothing, we regulate a curriculum for the model to generalize better over incremental tasks. This curriculum is built over *Orthogonal Weights Modification (OWM)* (Zeng et al., 2019) as the backbone in the downstream training. Orthogonal Weights Modification alleviates catastrophic forgetting by initiating weight updates along the orthogonal direction. Further, we employ a novel task-wise weight-regularizer that prevents information loss from the previous to the next incremental task. Penultimate Weight Sharing (PWS) prevents the transfer of pretext-specific knowledge to the downstream task. Through successive cycles of pre-training with SLA and downstream classification for each incremental task, we achieve state-of-the-art performance on the split-CIFAR10, split-CIFAR100, split-SVHN, and split-TinyImageNet databases.

The key highlights of the paper are summarized below:

- A novel self-supervised pretext task termed as *Stochastic Label Augmentation* (SLA) for learning rich and diverse feature representations.

- A curriculum-based learning technique for class-incremental learning through feature level smoothing.

- A novel regularization loss to constrain weight modification and prevent forgetting and Penultimate Weight Sharing (PWS) to prevent the transfer of pretext task-specific knowledge to the downstream task.

- We validate the significance of the curriculum algorithm by analyzing the shift-invariance progress and grad-cam visualizations.

- Evaluation of the proposed algorithm on four benchmark datasets: split-CIFAR10, split-CIFAR 100, split-SVHN, and split-TinyImageNet dataset and two high-resolution datasets: split-STL10 and ImageNet-100.

## 2 Related Work

The concept of incremental learning (also known as lifelong learning or continual learning) has been an active area of research in the deep learning community (Mai et al., 2022; Delange et al., 2021). Within the literature, three types of incremental learning scenarios have been explored i.e. domain-incremental learning, task-incremental learning, and class-incremental learning. The addition of new classes in an existing model (considered as an incremental task for the model) is referred to as class incremental learning (CIL). In this paper, we predominantly focus on the problem of CIL. With the addition of new classes, it is imperative that previously learned classes are not forgotten. This ties Incremental learning closely to the problem of catastrophic forgetting. To mitigate catastrophic forgetting, three popular classes of techniques exist, namely: (a) expansion based, (b) memory based, and (c) regularization based.

***Expansion based algorithms*** add new neurons (or parameters) that evolve with every task to allow the network to accumulate information of new classes sequentially. Rusu et al. (2016) introduced progressive neural networks in which modules with lateral connections are added with each task while preserving the base network. Dynamically Expandable Network Yoon et al. (2017) was proposed to competently calibrate the dynamic capacity of the network for sequential tasks. Li et al. (2019) isolated the neural architecture search framework and parameter tuning technique to identify the optimal structure for incremental tasks actively. Inspired by transfer learning, Sarwar et al. (2019) presented a clone and branch technique for efficient and dynamical adaptation in the incremental learning network. To alleviate model complexity, Yoon et al. (2019) introduced additive parameter decomposition, separating and tuning the network parameters as task-specific or task-shared.

***Memory-based models*** are either based on leveraging the subsets of the data from previous tasks (exemplars) or iteratively synthesizing the data based on the first task. Rebuffi et al. (2017) proposed iCaRL, which utilizes the exemplars from memory for rehearsal in continual learning. Deep generative replay framework (Shin et al., 2017) was introduced to sample data from the previous task and fuse it with data for the current task. Lopez-Paz & Ranzato (2017) implemented a Gradient Episodic Memory (GEM) model and applied loss gradients on current tasks to preserve information from previous tasks and prevent interference with memory. Average GEM (A-GEM) (Chaudhry et al., 2018) with altered loss function was presented as a more memory-efficient and better-performing variant of GEM. Riemer et al. (2018) addressed the trade-off between information transfer and interference by introducing a meta-experience replay algorithm to manage the transfer and interference based on future gradients. Distillation-based techniques which preserve knowledge from old classes through storing exemplars have also been proposed recently (Hou et al., 2019; Wu et al., 2019). In the recent research from Cha et al. (2021) and Ji et al. (2022), the authors proposed rehearsal-based techniques that preserve learned representations through a self-supervised distillation step.

In ***regularization-based techniques***, catastrophic forgetting is tackled by strategic regularization to support controlled weight updates based on previously learned parameters and the significance of past tasks. Elastic Weight Modification (Kirkpatrick et al., 2017) computes the importance of previous task weights and distribution of data based on the diagonal elements of the Fischer information matrix. Some work (Zenke et al., 2017; Aljundi et al., 2018) use appropriate synapses to efficiently accumulate and utilize relevant information from previous tasks to prevent catastrophic forgetting while learning new tasks. Ritter et al. (2018) apply Gaussian Laplace approximation of Hessian to estimate the task-based posterior. Farajtabar et al. (2020) update the new task weights orthogonally to the gradient direction of previous tasks. Subsequently, distillation methods (Hinton et al., 2015; Li & Hoiem, 2017; Hu et al., 2018) have also helped extract relevant information from previous tasks and impose regularization. The stability and plasticity

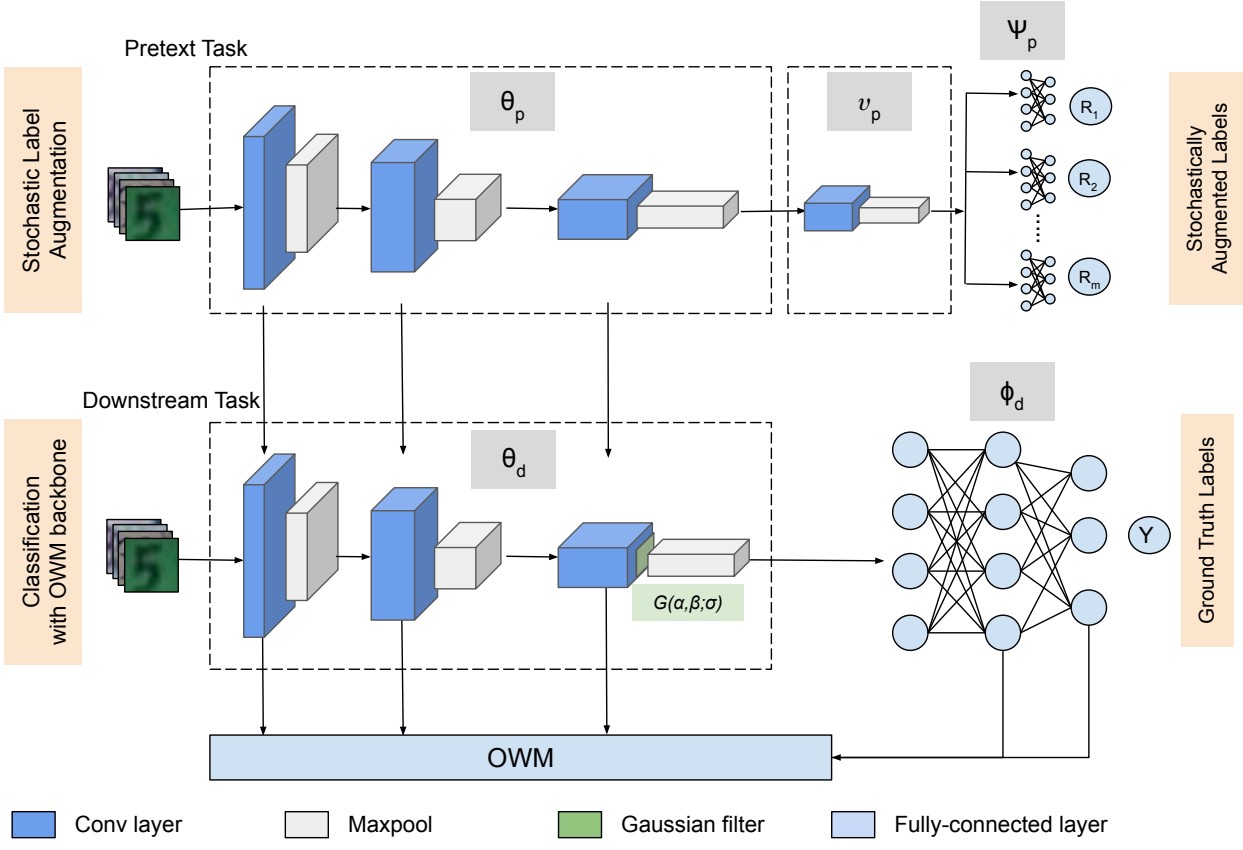

Figure 2: Block diagram of the proposed pretext and the downstream task for a particular incremental task $t$.

dilemma was addressed by framing two types of residual blocks in Adaptive Aggregation Networks (Liu et al., 2021). Combining knowledge distillation and replay, Boschini et al. (2022) introduced eXtended-DER (X-DER), where the model can revise the replay memory. Buzzega et al. (2020) took this idea a step further and combined knowledge distillation and replay with a regularization-based technique. CO-transport for class Incremental Learning (COIL) adapts to new tasks by learning the class-wise semantic relationship across incremental tasks (Zhou et al., 2021). In the study from Liu et al. (2020b), the authors proposed an ensemble of small classifiers coupled with regularization-based methods and achieved promising results. In a few recent research (Zhu et al., 2021a; Petit et al., 2023), the author proposed a memory-free generative approach where samples from previous tasks are generated and appended with the data available for the current incremental task.

## 3   Methodology

Let $D = \{D^t\}_{t=1}^{T}$ be the dataset organized for continual learning. Here $D^t$ consists of a set of $N^t$ images for the incremental task $t \in [1, T]$. Each incremental task $t$ corresponds to the addition of a new set of classes. The dataset $D$ constitutes $(x_i^t, y_i^t)^t$ where $x_i^t \in X^t$ represents the set of images per task and $y_i^t \in Y^t$ as the corresponding ground truth-label for task $t$. During each incremental task, we define a model $f$ with parameters $\theta^t$ and $\phi^t$ where $\theta$ denotes the parameters in the convolution layers and $\phi$ denotes the parameters in the fully-connected (FC) layers. The model is trained incrementally per task $t$ on $D^t$ thus culminating the final model as $f(\theta^T, \phi^T)$. Figure 2 illustrates the block diagram describing the framework for the pretext and downstream training for a single incremental task $t$. The proposed algorithm consists of:

1. *Unsupervised Pre-Training by Stochastic Label Augmentation:* In this step, we propose a novel pretext task for feature extraction. The pretext task augments labels instead of images for learning feature-rich representations.

2. *Downstream Training in Class-Incremental Setting:* For the downstream task, we employ a curriculum-based smoothing mechanism in combination with Orthogonal Weight Modification (OWM).

3. *Self-Supervised CIL with Task Regularization:* The model is trained iteratively for the pretext and downstream tasks with the proposed task regularization term to prevent catastrophic forgetting.

4. *Penultimate Weight Sharing:* The weights obtained after training the pretext model is transferred to the downstream model with the exception of weights from the last layer.

## 3.1 Unsupervised Pre-Training by Stochastic Label Augmentation

Continual learning models face the challenge of being unable to extract multiple significant features from images, unlike humans. Their performance is optimized based on the classification objective function, extracting only the features necessary for the current task. This can cause a loss of prior information, as the model is unaware of the features required for joint classification with previously learned tasks. For instance, if a model learns to classify dogs and birds by counting their legs, it may lack sufficient information to classify a cat in a subsequent task. Without replaying previous inputs, the model cannot extract additional information about previous classes as they are no longer available. We term this phenomenon as *Allocentric Ignorance*. As more tasks are added in a continual setting, the model becomes egocentric with current data at hand and selects only the features necessary for current and subsequent incremental tasks, potentially leading to missing parts that affect joint classification accuracy. To address the problem of allocentric ignorance, we employ self-supervised pre-training where model $f$ is first trained for a novel pretext task termed as *Stochastic Label Augmentation (SLA)*. In Self-Supervised Learning (SSL), we generally augment the input images during pretext tasks, which is computationally inefficient. Consequently, we propose *SLA*, which is based on augmenting the makeshift labels instead of images. These labels are sampled from a uniform distribution for $M$ tasks, each with $C$ classes. The training is performed in a multitask fashion with different fully-connected layers for each of the $M$ tasks (Figure 2). The major advantage of the proposed pretext task over the existing approaches (for instance, rotation pretext) is that the training time and resource usage do not increase significantly due to an increase in data ($4\times$ in the case of rotation pretext (Komodakis & Gidaris, 2018)).

We integrate self-supervised pre-training in CIL setting by training the network for a pretext task $p$ at each incremental task $t$. $f(\theta_p^t, \upsilon_p^t, \psi_p^t)$ represents the network for the pretext task $p$ where $\theta_p^t$ denotes the convolution parameters except the last block, $\upsilon_p^t$ denotes the last convolution block, and $\psi_p^t$ denotes the weights of the fully-connected layers for pretext task $p$ which gives the softmax of logits as output. The convolution weights $\theta_p^t$ (excluding $\upsilon_p^t$) are transferred to the downstream task $d$ for incremental task $t$. The model $f(\theta_p^t, \upsilon_p^t, \psi_p^t)$ utilizes only the images $X^t \in D^t$ for pre-training. Subsequently, $M$ branches of fully-connected layers ($\psi_p$) are added to the CNN corresponding to each of the $M$ tasks. This model is trained in through multitask learning where the features are extracted from the convolutional layers and then, for each of the $M$ tasks (not to be confused with the incremental task $t$), and fully-connected layers are trained with the cumulative loss incurred from all the tasks. The loss function used for training the network in pretext task is:

$$\min_{\theta_p^t, \upsilon_p^t, \psi_p^t} \mathbb{E}_{(x^t,y^t)\sim D^t} L(f(\theta_p^t, \upsilon_p^t, \psi_p^t; x^t), y^t) = \min_{\theta_p^t, \upsilon_p^t, \psi_p^t} \mathbb{E}_{(x^t,y^t)\sim D^t} \frac{1}{N^t} \sum_{i=1}^{N^t} \sum_{j=1}^{M} \sum_{k=1}^{C} -y_{i,j,k}^t log f(\theta_p^t, \upsilon_p^t, \psi_p^t; x_{i,j}^t) \quad (1)$$

where each $y_{i,j,k}$ is the stochastic label assigned to data point $x_{i,j}$. The model takes each image as input and predicts vectorized probabilities corresponding to each task. $L(.)$ is the cross-entropy loss minimized between the target vector of stochastically generated labels and the predicted vector. The pretext training is described in Algorithm 1.

In a regularization-based class-incremental setting, the model only learns discriminative features from images that are required for a single-incremental task. These features are not discriminative enough as more tasks are introduced to the model. Pre-training the model with unlabeled data drives the model to extract more and more information from

---

**Algorithm 1:** Pretext training using *SLA* for task $t$

---

**Input:** Images $X^t$, Downstream model conv-layer parameters $\theta_d^{t-1}$ from previous task $t-1$

**Initialize:** $\theta_d^0$ is zero-initialized.

**Parameters:** Number of epochs $E$, Number of stochastic tasks $M$, Number of classes per stochastic task $N$, Pretext model conv-layer parameters $\theta_p^t$, Pretext model FC parameters $\psi_p^t$, hyperparameters $a$ and $b$

**Function** *train_pretext_model($X^t, \theta_d^{t-1}$)*

    ***Initialize model*** $f(\theta_p^t, \upsilon_p^t, \psi_p^t)$

    $R^t = generate\_stochastic\_labels(X^t, m, n)$

    **for** *e=1 to E* **do**

        $\hat{R}^t = f(X^t; \theta_p^t, \upsilon_p^t, \psi_p^t)$

        ***Calculate loss terms:***

        $L_1 = \sum_{i=1}^M (\sum_{j=1}^N -R^t log(\hat{R}^t))$

        $L_2 = (a/2)\left\|\theta_p^t - \theta_d^{t-1}\right\|_2^2 + (b/2)\left\|\theta_p^t\right\|_2^2$

        $L = L_1 + L_2$

        ***Backpropagate loss*** $L$ ***and update*** $\theta_p^t$ ***and*** $\psi_p^t$

    **end**

    ***return*** $\theta_p^t$

**end**

---

each input image as the model will be forced to minimize the loss for each of the tasks. After pre-training the model $f(\theta_p^t, \upsilon_p^t, \psi_p^t)$ for the incremental task $t$ through the pretext task, Penultimate layer Weight Sharing (PWS) is adopted to transfer the convolution weights $\theta_p^t$ to the downstream model $f(\theta_d^t, \phi_d^t)$. The proposed pretext task learns generalized, unnoticed, and discreet features from the limited data and thus, save training time and computation resources.

## 3.2 Downstream Training in Class-Incremental Setting

The downstream task for classification in a class-incremental setting involves learning of the downstream model $f(\theta_d^t, \phi_d^t)$ for each incremental task $t$. For this learning, we propose a curriculum-based learning approach with an OWM backbone (Zeng et al., 2019). The downstream training of the model using these components is described in Algorithm 2.

### 3.2.1 Orthogonal Weight Modification (OWM)

The OWM technique has been shown to address the problem of catastrophic forgetting commonly observed in continual learning problems (Zeng et al., 2019). In the OWM technique, an orthogonal projection matrix $P_l$ is considered on the input space of layer $l$. This projector is defined as

$$P_l = I - A_l(A_l^T A_l + \alpha I)^{-1} A_l^T \tag{2}$$

where $\alpha$ is a constant to calculate the inverse of the matrix and $I$ is a unit matrix. Matrix $A_l$ consists of all trained input vectors spanning the input space where the previous task has already been learned as its columns, e.g., $A_l = [a_1, \ldots, a_{l-1}]$. For each incremental task $t$, the weights $\phi_l^t$ and projector $P_l$ are updated for each layer $l$ in the network, such that the information learned in the previous tasks is retained. The weights $\phi_l^t$ are updated as:

$$\Delta\phi_l^t = \lambda P_l^{t-1} \Delta\phi_l^{t\,(BP)} \tag{3}$$

where $\lambda$ is the learning rate, $\Delta\phi^{(BP)}$ is the standard weight update using the backpropagation algorithm. The projector P may be updated using an iterative or a recursive method to obtain a correlation-inverse matrix (Shah et al., 1992; Haykin, 2008). We use OWM as the backbone technique to address the problem of catastrophic forgetting in the proposed algorithm.

---

**Algorithm 2:** Downstream learning algorithm for task $t$

---

**Input:** Images $X^t$, Labels $Y^t$, Pretext model conv-layer parameters $\theta_p^t$ for current task $t$
**Initialize:** $\theta_d^t$ is initialized with $\theta_p^t$.
**Parameters:** Number of epochs $E$, Downstream model FC parameters $\phi_d^t$, Gaussian filter $G$ with standard
  deviation $\sigma$, constant $c$
**Function** *train_downstream_model($X^t, Y^t, \theta_d^t$)*
    *Initialize model* $f(\theta_d^t, \phi_d^t)$
    *Set* $\sigma_e$ *= 1*
    **for** *e=1 to E* **do**
        $\hat{a}_n = f(X^t; \theta_d^t, \phi_d^t)$
        $\sigma_e$ *=* $\sigma_e \cdot c$
        $z_{n+1} = pool(G(\sigma_e) * \hat{a}_n)$
        $\hat{Y}^t = argmax(z_{n+1})$
        *Calculate loss terms:*
        $L = \sum_{i=1}^{M}(\sum_{j=1}^{N} -Y^t log(\hat{Y}^t))$
        *Backpropagate loss* $L$ *and update* $\theta_d^t$ *and* $\phi_d^t$ *using OWM algorithm.*
    **end**
    *return* $\theta_d^t$, $\phi_d^t$
**end**

---

### 3.2.2 Smoothing-based Curriculum Learning

We design a curriculum-based learning technique for training the downstream model. Recent work has shown the effectiveness of Gaussian smoothing in the context of curriculum learning (Chen et al., 2019; Sinha et al., 2020). Convolution of a conventional smoothing kernel with an input signal results in a blurring effect. This means that some information in the input is lost. In other words, the smoothing kernel regulates the information that is propagated after each convolution operation.

A Gaussian filter $G$, parameterized with $\sigma_e$ is applied to the extracted feature maps from the last convolution layers of the model. While training over $e$ epochs, we increase the strength of the smoothing filter $G$ simply by increasing $\sigma_e$. In a standard CNN model with weights, $\theta_l$ the following operations are performed at layer $l$. The activations at each layer is obtained as $\hat{a}_l = ReLU(\theta_l * z_l)$ followed by pooling, described as $z_{l+1} = pool(\hat{a}_l)$. Here $\hat{a}_l$ denotes the activated output using the rectified linear unit $ReLU$, input $z_l$ denotes the input at layer $l$, $*$ is the convolution operation and $pool$ is the max-pooling layer. For an $n$-layer CNN with weights $\theta_l$ at each layer $l$, we integrate a smoothing filter after the $n^{th}$ convolution layer of the CNN. This can be expressed as follows:

$$z_{n+1} = pool(G(\sigma_e) * \hat{a}_n) \tag{4}$$

where $z_{n+1}$ becomes the input to the first fully-connected layer.

The curriculum defined above is built over the observation that for each incremental task $t$, the feature maps obtained at the last convolutional layer have an abundance of information from which a model can learn. Plenty of information in the maps allows the model to focus on the features that are easy to extract and lead to the best optimization of the objective. This makes it an easy sample in the curriculum. Over the epochs, the difficulty of the curriculum is increased by repressing the high-frequency information from the feature maps. This is achieved by increasing the standard deviation of the smoothing kernel. The model is then forced to extract discriminative, inconspicuous, or obscured features from the smoothened feature map. This provides a curriculum-based training where the model learns to classify with a lesser and lesser amount of information. Furthermore, Gaussian smoothing filters have traditionally been used for anti-aliasing in image processing (Gonzalez, 2009). Anti-aliasing, when integrated correctly, has been shown to enhance the shift-invariance tendency of CNNs (Zhang, 2019). The fusion of smoothing (blur) filters with pooling/strided convolution softens the feature maps and alleviates the variance introduced by operations that predominantly ignore the Nyquist sampling theorem. Recently, the fusion of these filters has shown a boost in the performance and generalization capacity of the CNN models Zhang (2019); Zou et al. (2020). By incorporating a

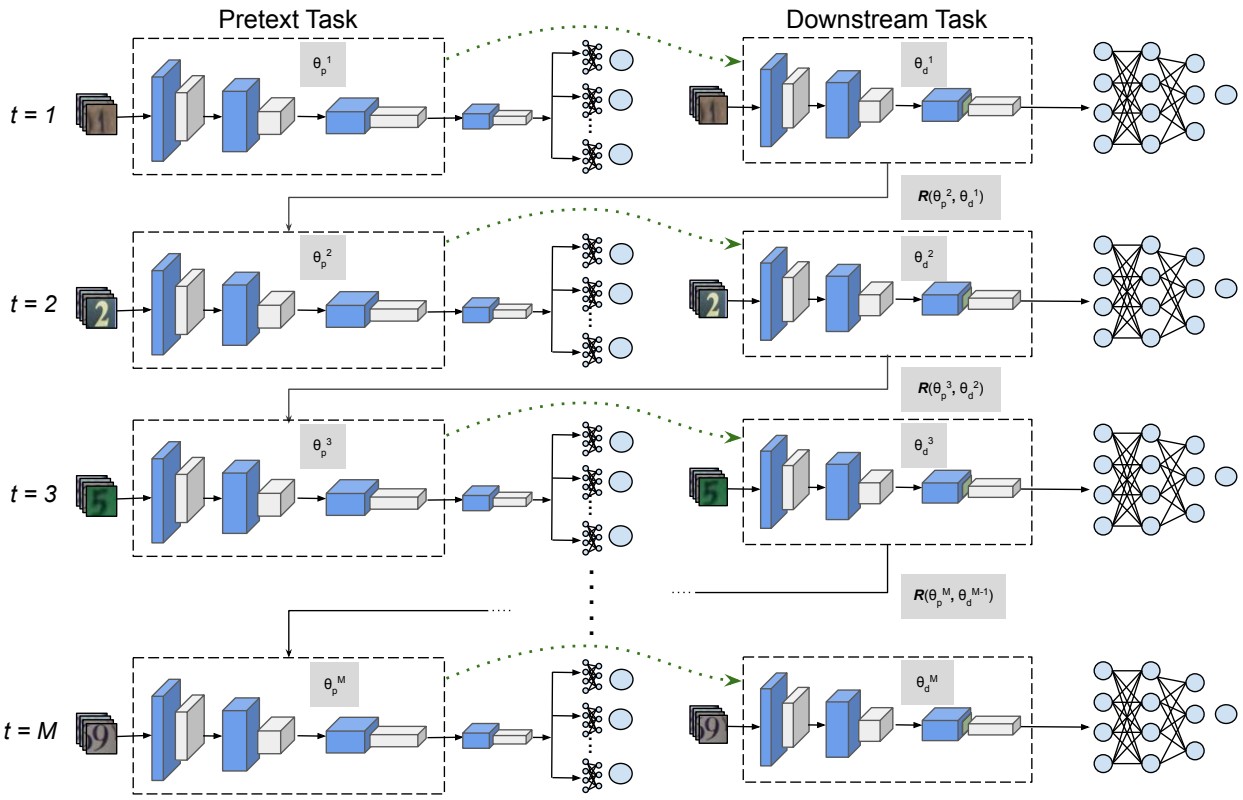

Figure 3: Block diagram to demonstrate the complete training procedure of the proposed algorithm for $M$ incremental tasks.

Gaussian smoothing-based curriculum for training, we expect improved robustness of the model towards a shift in input.

### 3.3 S$^2$C$^2$IL: Self-Supervised Curriculum-based Incremental Learning

Since the model is being trained in a class-incremental fashion, we only have limited data corresponding to the classes that are introduced. To make maximum use of the limited amount of data, we begin with the pretext task of self-supervision using the data at hand. This ensures that the model is able to learn good feature representations from the provided data. In the next step, the model learns to perform classification between the given set of classes using OWM combined with curriculum learning.

The cycle of pre-training and downstream classification is repeated every time a new set of classes arrive (Figure 3). In practical instances, it is highly likely that the dataset associated with the new incremental task may belong to an entirely different distribution. Training the existing model on this new, out-of-distribution dataset may lead to excessive modification of convolution weights, and the final model may fail to generalize on the old incremental tasks. To prevent forgetting for convolutional layers at each incremental step, we incorporate a regularization term in the calculated loss Xuhong et al. (2018). The regularization term ensures that the weights are not drastically modified after each pre-training and classification cycle. The complete S$^2$C$^2$IL algorithm is presented in Algorithm 3. Mathematically put, while training the model for the pretext task of incremental task $t-1$ on dataset $D^{t-1}$, we transfer the convolution weights $\theta_p^{t-1}$ from pretext model $f(\theta_p^{t-1}, \psi_p^{t-1})$ to downstream model $f(\theta_d^{t-1}, \phi_d^{t-1})$. After this, on the introduction of the next incremental task $t$, the model is first trained for the pretext task. For this, the weights from the previous downstream task $\theta_d^{t-1}$ are transferred. We add a weight regularization term between the convolution weights between $\theta_d^{t-1}$ of the previous task and $\theta_p^t$ of the current task to mitigate forgetting at this step. In other words, we incur a regularization loss term $R$ between the model trained on the downstream task of incremental task $t-1$ and the pretext

---

**Algorithm 3:** $S^2C^2IL$ algorithm

---

**Input:** Total tasks $T$, Images $X$, Labels $Y$

**Parameters:** Pretext model conv-layer parameters $\theta_p$, Pretext model FC parameters $\psi_p$, Downstream model conv-layer parameters $\theta_d$, Downstream model FC parameters $\phi_d$

**for** $t$ = 1 to $T$ **do**

$\quad \theta_p^t = train\_pretext\_model(X_t, \theta_d^{t-1})$                                                 //Algorithm 1

$\quad \theta_d^t, \phi_d^t = train\_downstream\_model(X_t, Y_t, \theta_p^t)$                              //Algorithm 2

**end**

Evaluate model $f(\theta_d^T, \phi_d^T)$ trained for $T$ tasks.

---

task for the model being trained on the next incremental task i.e. incremental task $t$. The proposed regularization loss incurred on two consecutive incremental tasks is:

$$\boldsymbol{R}(\theta_p^t, \theta_d^{t-1}) = \frac{\boldsymbol{a}}{2} \left\| \theta_p^t - \theta_d^{t-1} \right\|_2^2 + \frac{\boldsymbol{b}}{2} \left\| \theta_p^t \right\|_2^2 \tag{5}$$

where $a$ and $b$ are hyperparameters, and $\boldsymbol{R}(.)$ is the regularization loss optimized with the standard multitask cross-entropy loss. The hyperparameter $a$ is a constant that aggravates the loss, forcing the model not to deviate much from the model trained on the previous task. The hyperparameter $b$ handles the induced sparsity on the model being trained on the current incremental task $t$.

## 4 Experimental Setup

The proposed algorithm is evaluated on four benchmark datasets: split-CIFAR10, split-CIFAR100, split-SVHN, and split-TinyImageNet and two high-resolution datasets: split-STL10 and ImageNet-100. We report the average test accuracy, which is defined as the average of test accuracies achieved across all tasks. All experiments are performed using five fixed random seeds. The proposed algorithm is evaluated under two settings- (i) OWM + CL, and (ii) $S^2C^2IL$. In the first setting, only the curriculum-based downstream model is trained without any self-supervision. In the $S^2C^2IL$ setting, we follow the methodology as explained in Section 3.3, and perform pre-training using the proposed SLA technique.

**Datasets and Protocol:** Since the focus of this work is class-incremental setting, we train and test the proposed algorithm according to the protocols defined in the works of Zeng et al. (2019) and Hu et al. (2018). For experiments, we have used six datasets:

(i) **Split-CIFAR10 (Krizhevsky, 2009)** contains 60,000 $32 \times 32$ color images of 10 different classes with 50,000 images in the training set and 10,000 images in the testing set. The training and evaluation is performed for 2 classes per task.

(ii) **Split-CIFAR100 (Krizhevsky, 2009)** contains 60,000 $32 \times 32$ color images of 10 different classes with 50,000 images in the training set and 10,000 images in the testing set. The training and evaluation are done for 10, 20, and 50 classes per task.

(iii) **Split-SVHN (Netzer et al., 2011)** contains 60,000 $32 \times 32$ color images of 10 different classes with 50,000 images in the training set and 10,000 images in the testing set. The training and evaluation are performed for 2 classes per task.

(iv) **Split-TinyImageNet (Le & Yang, 2015)** contains 120,000 color images of size $64 \times 64$ from 200 different classes with 100,000 images in the training set, 10,000 in the validation set and 10,000 images in the testing set. The training and evaluation of the model are done for 5, 10, and 20 classes per task.

(v) **Split-STL10 (Coates et al., 2011)** contains 13,000 color images of size $96 \times 96$ from 10 classes with 5000 images in the training set, 8000 in the testing set. The training and testing of the model are done for 2 classes per task.

| Methods | Split-CIFAR10 | Split-CIFAR100 | | | Split-SVHN |
|---|---|---|---|---|---|
| | 5 tasks | 2 tasks | 5 tasks | 10 tasks | 5 tasks |
| EWC* (Kirkpatrick et al., 2017) | $31.40 \pm 2.21$ | $27.58 \pm 1.64$ | $18.42 \pm 1.53$ | $13.28 \pm 0.91$ | $34.22 \pm 3.83$ |
| iCaRL* (Rebuffi et al., 2017) | $50.02 \pm 2.04$ | $24.20 \pm 1.60$ | $22.16 \pm 0.86$ | $19.00 \pm 0.36$ | $71.25 \pm 0.67$ |
| PGMA (Hu et al., 2018) | $40.47$ | - | - | - | - |
| DGM* (Ostapenko et al., 2019) | $50.53 \pm 0.46$ | $28.23 \pm 0.75$ | $25.43 \pm 0.14$ | $24.09 \pm 0.19$ | $73.01 \pm 0.77$ |
| OWM (Zeng et al., 2019) | $55.71 \pm 0.49$ | $40.30 \pm 0.65$ | $33.17 \pm 0.79$ | $29.86 \pm 0.33$ | $73.50 \pm 0.81$ |
| MUC* (Liu et al., 2020b) | - | $33.86 \pm 0.72$ | $28.05 \pm 1.22$ | $22.07 \pm 0.9$ | - |
| IL2A* (Zhu et al., 2021a) | - | $\underline{43.29 \pm 0.43}$ | $32.63 \pm 0.86$ | $21.45 \pm 0.67$ | - |
| PASS* (Zhu et al., 2021b) | - | $43.15 \pm 0.31$ | $34.89 \pm 0.75$ | $24.03 \pm 0.74$ | - |
| SSRE* (Zhu et al., 2022) | - | $41.06 \pm 0.87$ | $\mathbf{36.82 \pm 0.7}$ | $31.35 \pm 1.0$ | - |
| FeTrIL* (Petit et al., 2023) | - | $40.88 \pm 1.18$ | $35.47 \pm 1.15$ | $\mathbf{32.50 \pm 1.03}$ | - |
| **OWM + CL (Ours)** | $\underline{58.68 \pm 0.37}$ | $43.10 \pm 0.66$ | $35.40 \pm 0.36$ | $31.37 \pm 0.61$ | $\underline{75.34 \pm 0.64}$ |
| **S²C²IL (Ours)** | $\mathbf{61.64 \pm 0.57}$ | $\mathbf{43.98 \pm 0.65}$ | $\underline{35.59 \pm 0.49}$ | $\underline{31.93 \pm 0.54}$ | $\mathbf{77.53 \pm 0.53}$ |

Table 1: Average test accuracy for proposed method on Split-CIFAR10, Split-CIFAR100, and Split-SVHN dataset. The best performance is depicted by **bold** and the second best by underline. All results are cited from Kirkpatrick et al. (2017); Rebuffi et al. (2017); Ostapenko et al. (2019); Hu et al. (2018); Zeng et al. (2019); Liu et al. (2020b); Zhu et al. (2021a;b; 2022); Petit et al. (2023) or are reproduced from their official repository for a fair comparison ($*$ means re-run with protocols described in this paper)

| Methods | 5 tasks | 10 tasks | 20 tasks |
|---|---|---|---|
| OWM (Zeng et al., 2019) | $19.00 \pm 0.28$ | $16.05 \pm 0.27$ | $14.30 \pm 0.32$ |
| SLA + OWM | $20.59 \pm 0.32$ | $17.05 \pm 0.58$ | $15.08 \pm 0.65$ |
| OWM + CL | $21.12 \pm 0.42$ | $17.56 \pm 0.33$ | $15.54 \pm 0.12$ |
| S²C²IL | $\mathbf{21.39 \pm 0.15}$ | $\mathbf{19.00 \pm 0.35}$ | $\mathbf{19.52 \pm 1.56}$ |

Table 2: Average accuracy (%) reported for the ablation experiments performed on the split-TinyImageNet dataset for 5, 10, and 20 tasks.

(iv) **ImageNet-100 (Russakovsky et al., 2015)** is a subset of ImageNet ILSVRC 2012 (Russakovsky et al., 2015) dataset with 100 classes. In this, the training set consists of 1300 images per class and the validation set consists of 50 images per class. Images are set to a size of $224 \times 224$. The training and evaluation of the model are done only for 10 classes per task.

**Comparitive Algorithms:** The results of the proposed framework are compared with various benchmark algorithms in the domain of regularization-based CIL with the exception of iCaRL. The following algorithms are used for comparison: (1) EWC (Kirkpatrick et al., 2017), (2) iCaRL (Rebuffi et al., 2017) with 2000 exemplars; (3) PGMA (Hu et al., 2018); (4) DGM (Ostapenko et al., 2019), (5) OWM (Zeng et al., 2019), (6) MUC (Liu et al., 2020b), (7) IL2A (Zhu et al., 2021a), (8) PASS (Zhu et al., 2021b), (9) SSRE (Zhu et al., 2022), and FeTrIL (Petit et al., 2023). The EWC[1], iCaRL[1], DGM[2], OWM[3], MUC[4], IL2A[5], PASS[6], SSRE[7], and FeTrIL[8] baselines are run using open-source codes with the same network architecture as the one used in S²C²IL. The details of this network are described in Section 4. Further, S²C²IL is compared with various memory-based approaches on the Split-TinyImageNet dataset. It should be noted that the proposed S²C²IL algorithm uses *no* exemplars from classes of previous tasks.

---

[1] https://github.com/mmasana/FACIL

[2] https://github.com/SAP-archive/machine-learning-dgm

[3] https://github.com/beijixiong3510/OWM

[4] https://github.com/liuyudut/MUC

[5] https://github.com/Impression2805/IL2A

[6] https://github.com/Impression2805/CVPR21$_P ASS$

[7] https://github.com/zhukaii/SSRE/tree/5475c9803b0143cab849b62edb7d5db76433c388

[8] https://github.com/G-U-N/PyCIL

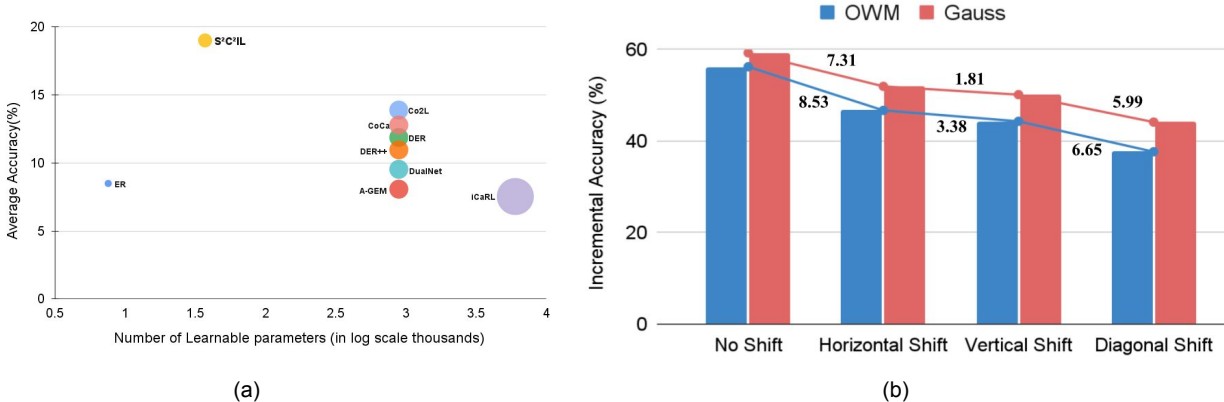

(a)             (b)

Figure 4: (a) Comparison of the proposed $S^2C^2IL$ algorithm with memory-based continual learning algorithms (Riemer et al., 2018; Chaudhry et al., 2019; Rebuffi et al., 2017; Benjamin et al., 2018; Buzzega et al., 2020; Pham et al., 2021; Cha et al., 2021; Ji et al., 2022). The accuracy achieved by each algorithm on the split-TinyImageNet dataset (for 10 incremental tasks) is plotted against the number of convolution parameters (in log scale). The size of each bubble corresponds to the network size used by the algorithm. The results obtained are summarized in Table 4. (b) Bar plot demonstrating the incremental accuracies for OWM and Gaussian-based OWM model when trained on split-CIFAR10 dataset for 5 tasks. The line graph summarizes the subsequent drop in the accuracy caused by pixel translations. Gaussian network displays far less performance drop for translated dataset than base OWM model.

**Implementation Details:** For all the experiments, we use a 3-layer CNN network with three fully-connected layers. The same network architecture is used by Zeng et al. (2019). For each incremental task, we start the model training on the pretext task. Here, we use the 3-layer CNN architecture for feature extraction and utilize these features in multitask fashion. For our experiments, we train the model for three tasks with two classes each, i.e., the extracted features are utilized by three separate heads of fully-connected layers with two layers each. For the downstream task, the same weights from the pretext task are transferred. However, here the features are utilized by a single fully-connected layer to learn the current incremental task. The mentioned architectures used in the pretext and downstream can be better visualized in Figure 2. We train all the models with stochastic gradient descent (SGD). For the pretext task, the multitask network is trained on stochastically generated labels for three tasks with two classes each. We set the learning rate to 0.001 to train it for 50 epochs. The hyperparameters $a$ and $b$ are fixed to 10 and 18 for split-CIFAR10 and split-CIFAR100 datasets and 5 and 12 for split-SVHN datasets, respectively. As described in section 3, the model is trained for a curriculum where the training starts with $\sigma$ set to 0.9 with a decay rate of 0.95 for every 10 epochs for split-CIFAR100 and split-SVHN datasets. For the split-CIFAR10 dataset, $\sigma$ is set to 1 with a decay rate of 0.9 for every ten epochs. All experiments are performed for five random seeds and the performance is reported as the mean and standard deviation over all the seeds. The algorithm is implemented in Pytorch, and all the experiments are performed on a DGX station with 256 GB RAM and four 32 GB Nvidia V100 GPUs. For reproducibility, the source code will be released in the camera-ready version.

| Pretext Task | Average Accuracy (%) | |
|---|---|---|
| | **split-CIFAR-10** | **split-SVHN** |
| Without Pre-training | $55.71 \pm 0.49$ | $73.50 \pm 0.81$ |
| Rotation (Komodakis & Gidaris, 2018) | $57.59 \pm 0.43$ | $76.22 \pm 0.23$ |
| Colorization (Larsson et al., 2017) | $56.66 \pm 0.92$ | $76.10 \pm 0.69$ |
| SLA (proposed) | $\mathbf{61.64 \pm 0.57}$ | $\mathbf{77.53 \pm 0.53}$ |

Table 3: Performance of the proposed algorithm by replacing the proposed Stochastic Label Augmentation (SLA) with Rotation and Image Colorization pretext tasks on the split-CIFAR10 and split-SVHN datasets for 5 tasks.

# 5 Results and Analysis

The performance of the proposed $S^2C^2IL$ algorithm on the split-CIFAR10, split-CIFAR100, and split-SVHN datasets are reported in Table 1. From Table 1, it is observed that the proposed algorithm achieves state-of-the-art performance on the split-CIFAR10 dataset and split-SVHN dataset when compared to the existing algorithms. When compared to the backbone algorithm (Zeng et al., 2019), $S^2C^2IL$ improves the average accuracy with up to 4% and 6% performance gain for the split-SVHN and split-CIFAR10 datasets, respectively. Further, we observe that training without self-supervision in $S^2C^2IL$ (OWM + CL) also outperforms existing algorithms on both datasets. For the split-CIFAT100 dataset, our proposed algorithm achieves state-of-the-art performance closely followed by IL2A (Zhu et al., 2021a) by a difference of 0.69% for 2 incremental tasks. $S^2C^2IL$ closely follows SSRE (Zhu et al., 2022) by 1.13% for 5 incremental tasks, and FeTrIL (Petit et al., 2023) by 0.57% for 10 incremental tasks.

In Table 2, we perform the ablation experiments and report the performance on the split-TinyImageNet dataset. The evaluation is performed for 5, 10, and 20 tasks. From Table 2, it can be observed how each component contributes towards mitigating catastrophic forgetting. We experimentally compare the proposed regularization-based $S^2C^2IL$ algorithm with a recent and state-of-the-art memory-based algorithm. In Figure 4 (a) the accuracy achieved by each algorithm is plotted against the number of parameters in convolution layers (in log scale of thousands). It can be clearly visualized that the proposed $S^2C^2IL$ algorithm outperforms existing memory-based algorithms by a large margin on the split-TinyImageNet dataset for ten tasks. Further, the $S^2C^2IL$ framework exceeds the performance of memory-based algorithms even with a smaller backbone and *without* using any memory. The results obtained are summarized in Table 4. The results from Table 1 and Table 2 show the generalizability of the proposed $S^2C^2IL$ on various datasets showcasing the efficacy of the proposed algorithm.

To qualitatively evaluate the performance of the proposed algorithm, we employ GradCAMs. In Figure 5, the GradCAM visualization obtained after each incremental task using OWM algorithm and the proposed $S^2C^2IL$ algorithm is presented. We use the images from the first incremental task of the split-SVHN dataset. From the generated maps, it is observed that after each incremental task/step, the focus of the model diverges in the case of the OWM algorithm. However, the maps generated through $S^2C^2IL$ are better at retaining focus even after multiple incremental training steps. This highlights the stability of the proposed $S^2C^2IL$ algorithm and its effectiveness at delaying forgetting in the network.

**Efficacy of Stochastic Label Augmentation (SLA):** The proposed SLA task uses stochastically sampled labels, which are not semantically interpretable for humans per se. The pioneering work of Zhang et al. (2017) established that the models could capture meaningful information out of random labels even when there is no correlation between images and labels. Further research (Misra & Maaten, 2020) has shown that, for a given pretext task, the initial layers of a model learn generalized semantic features, while the final layers learn task-specific features. Recent studies on

| Algorithm | Accuracy (%) | Network Used | Parameters (in million) |
|---|---|---|---|
| iCaRL (Rebuffi et al., 2017) | $7.53 \pm 0.79$ | ResNet32 | 60 |
| ER (Riemer et al., 2018) | $8.49 \pm 0.16$ | **Four-layerd CNN** | **0.08** |
| A-GEM (Chaudhry et al., 2018) | $8.07 \pm 0.08$ | ResNet18 | 8.98 |
| DER (Buzzega et al., 2020) | $11.87 \pm 0.78$ | ResNet18 | 8.98 |
| DER++ (Buzzega et al., 2020) | $10.96 \pm 1.17$ | ResNet18 | 8.98 |
| DualNet (Pham et al., 2021) | $9.53 \pm 0.53$ | ResNet18 | 8.98 |
| Co2L (Cha et al., 2021) | $\underline{13.88 \pm 0.42}$ | ResNet18 | 8.98 |
| CoCa (Ji et al., 2022) | $12.78 \pm 0.0$ | ResNet18 | 8.98 |
| **$S^2C^2IL$ (Proposed)** | **$19.00 \pm 0.35$** | Four-layerd CNN | $\underline{0.37}$ |

Table 4: The accuracy achieved by different memory-based algorithms with backbone architecture used by each algorithm on the split-TinyImageNet dataset for 10 tasks. The best performance is depicted by **bold** and the second best by underline. The proposed $S^2C^2IL$ achieves the highest performance with no memory and with significantly smaller backbone architectures.

| Number of Tasks (M) | Performance (in %) |
|---|---|
| 1 | $59.93 \pm 0.42$ |
| 2 | $60.82 \pm 0.72$ |
| 3 | $61.64 \pm 0.57$ |

Table 5: The demonstration of the effect of increasing the number of tasks (varying values of M) in pretraining step of SLA on the performance of the proposed when trained on split-CIFAR10 for 5 incremental tasks.

training models with randomly labeled data have (Maennel et al., 2020) revealed that the generalized information learned leads to a positive transfer of features. At the same time, specialization in later layers results in a negative transfer from pretext to downstream tasks.

Inspired by the existing literature, we have proposed an SLA-based pretext task in which the initial layers of the model extract generalized information and the final layer learns specialized information. To facilitate the positive transfer of knowledge from pretext to downstream tasks while preventing negative transfer, we employ Penultimate Weight Sharing (PWS), which skips the transfer of weights from the final section of the model. Overall, our approach aims to extract as much information as possible from each image (Zhang et al., 2017) and transfer only the generalized information to downstream tasks, excluding any task-specific information that may have been learned in the process.

To understand the effectiveness of the proposed pretext task, we perform additional experiments on the split-CIFAR10 and split-SVHN datasets for five incremental tasks each. The proposed framework is tested after replacing the SLA pretext task with two existing pretext tasks, namely Rotation (Komodakis & Gidaris, 2018) and Image Colorization (Larsson et al., 2017). From Table 3, we observe that in comparison to Image Colorization and Rotation, SLA leads to a higher performance gain. We conduct t-tests to verify that the improvement on the split-CIFAR10 and split-SVHN datasets is statistically significant. The results show that a p-value of 0.0006 and 0.0013 is achieved when SLA is compared to Rotation and Colorization, respectively, on the split-CIFAR10 dataset. Similarly, on the split-SVHN dataset, a p-value of 0.0171 and 0.0465 is achieved when SLA is compared with Rotation and Colorization, respectively. The statistical test shows that the difference in the performance of different pretext tasks at a confidence score of 0.05 is statistically significant. In addition to performance gain through SLA, it should be noted that augmenting the labels instead of data leads to faster pre-training, lesser computational cycles, and low memory usage. The reason behind successful learning through augmented labels instead of data is due to the fact that a multitask network learns from the synergy of multiple tasks that it has to learn. Since the network is bound to minimize the loss, it will excerpt all the discriminating features to minimize it. Moreover, since the deep learning models are highly non-linear, they can reasonably achieve near-perfect accuracy on the training dataset (Zhang et al., 2017). The combination of these prospects leads to performance gain, which makes learning without the availability of true annotated labels possible.

**Effect of Multitask Learning in SLA pretext:** The study from the authors of Zhang et al. (2017) demonstrates that models can extract meaningful information from images, even when presented with random labels. Building on this finding, we extended the SLA to a multitask regime by augmenting each image with additional labels, thereby encouraging the model to extract even more information. As discussed in the literature (Malhotra et al., 2022), a multitask learning setting can be more effective than a single-task setting, as it provides more supervision for each image and allows the model to learn from different tasks in synergy. The empirical results in Table 5 support the existing literature, showcasing an improvement in performance when scaling the model from single tasks to multiple tasks in SLA-based pretraining.

| Technique | Time Taken (in seconds) |
|---|---|
| Rotation Pretext | 9 s |
| Colorization | 8 s |
| SLA | 2 s |

Table 6: Time taken (in seconds) for an epoch by different pretext tasks on split-CIFAR100 dataset for 5 incremental tasks.

| Dataset | Tasks | S$^2$C$^2$IL (w/o PWS) | S$^2$C$^2$IL |
|---|---|---|---|
| split-SVHN | 5 | $75.88 \pm 0.45$ | $77.53 \pm 0.53$ |
| split-CIFAR10 | 5 | $60.75 \pm 0.53$ | $61.64 \pm 0.57$ |
| | 2 | $43.38 \pm 0.20$ | $43.98 \pm 0.65$ |
| split-CIFAR100 | 5 | $35.52 \pm 0.48$ | $35.59 \pm 0.49$ |
| | 10 | $31.85 \pm 0.45$ | $31.93 \pm 0.54$ |

Table 7: Performance comparison of the proposed algorithm without and with Penultimate Weight Sharing (PWS). The proposed Stochastic Label Augmentation (SLA) task is used for unsupervised pre-training.

**Computational Time Requirements of the SLA and S$^2$C$^2$IL:** We compute the time taken by the proposed SLA algorithm and compare it with existing SSL techniques. Table 6 showcase that the proposed SLA algorithm takes significantly less time when compared with existing pretraining algorithms on split-CIFAR100 dataset for 5 incremental tasks. We also compute the training time of the existing algorithms (available in Table 1 for an epoch on split-CIFAR100 dataset for 5 incremental tasks. We observe that the existing algorithms, such as MUC (Liu et al., 2020a) and SSRE (Zhu et al., 2022) require about 8 seconds, and IL2A (Zhu et al., 2021a) and FeTrIL (Petit et al., 2023) require over 200 seconds to train for a single epoch. The proposed S$^2$C$^2$IL algorithm takes 6 seconds for each epoch, making it computationally efficient when compared to most of the existing baseline algorithms.

**Impact of Penultimate Weight Sharing (PWS):** Conventionally, the optimized weights of the pretext task are utilized for training the model on the downstream task. The quality of the pre-trained features consistently improves with the position and depth of layers. Further, the task accuracy is influenced by pre-training a network only up to $k$-layers (Misra & Maaten, 2020). We hypothesize that the deeper layers of the pretext task are calibrated toward the pretext task. To alleviate the bias towards the pretext task, we transfer weights from all layers except those of the last layer for fine-tuning on the downstream task. This weight transfer algorithm is termed Penultimate Weight Sharing (PWS). PWS incorporates layers weight ($\theta_p^t$) sharing for downstream fine-tuning (dropping out $\upsilon_p^t$) and empowers the network to learn generalized feature representations for task $t$. The benefit of sharing weights only up till the penultimate layer prevents sharing of pretext-specific weights to the downstream model.

To evaluate the model's performance in the absence of PWS, we remove the $\upsilon_p^t$ convolution block from the pretext model, rendering it equivalent to the downstream model's architecture. After pre-training the model and transferring the weights to the downstream model, the performance is evaluated on different datasets. The results presented in Table 7 highlight the performance improvements obtained by transferring weights up to the penultimate layer.

**Anti-aliasing Filters and Shift-Invariance:** In this work, we employ a Gaussian filter for smoothing the feature maps during the downstream classification task. Since the fusion of these filters has shown improved generalization capabilities in CNNs (Zhang, 2019; Zou et al., 2020), we study the impact of the filtering integrated with a downstream model by evaluating the performance of S$^2$C$^2$IL using these filters. We highlight the relevance of primitive integration of the filter by stacking it after a convolution block and studying shift-invariance properties through related performance metrics. Figure 4 (b) depicts the comparison of the incremental accuracy when a shifted/translated image is given as input to the OWM and Gaussian-based OWM network. We modify the split-CIFAR10 dataset by incorporating the random affine translation of 1% to ensure horizontal, vertical, and diagonal pixel shifts. The accuracies obtained

| | Average Incremental Accuracy | | | |
|---|---|---|---|---|
| | OWM | $\delta_{owm}$ | Gauss | $\delta_{Gauss}$ |
| Original | 56.2 | 0 | 59.23 | 0 |
| Horizontal Shift | 47.67 | 8.53 | 51.92 | $\mathbf{7.31}_{-1.22}$ |
| Vertical Shift | 44.29 | 11.91 | 50.11 | $\mathbf{9.12}_{-2.79}$ |
| 2D Shift | 37.64 | 18.56 | 44.12 | $\mathbf{15.11}_{-3.45}$ |

Table 8: Average accuracy (in %) response of the network to the original and shifted datasets. $\delta_{owm}$ and $\delta_{Gauss}$ represent the accuracy drop in the OWM and Gaussian-based OWM models. A higher value of $\delta$ implies a more adverse effect of shift on the model performance.

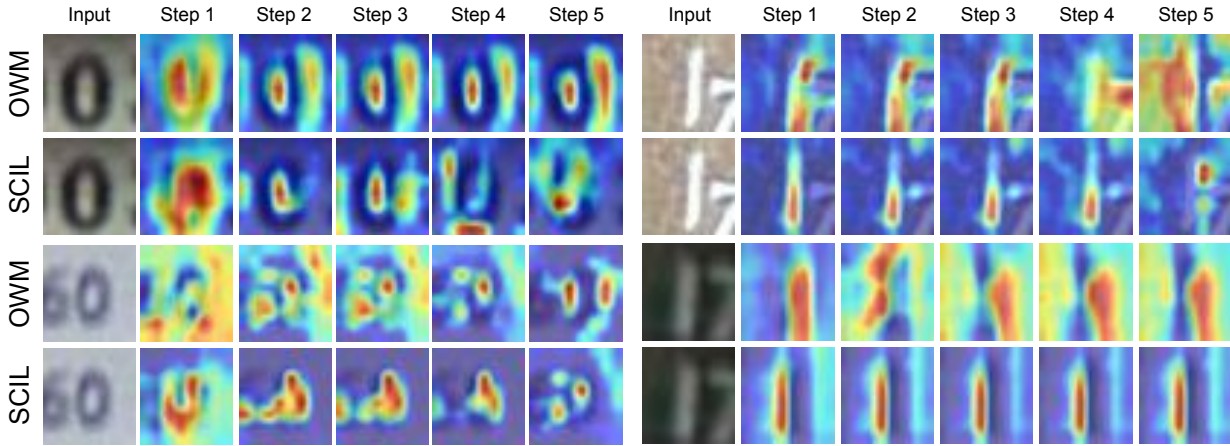

Figure 5: Task-wise GradCAM visualization (step in the figure stands for an incremental task) for the split-SVHN dataset using the OWM (row 1 and row 3) and proposed $S^2C^2IL$ algorithm (row 2 and row 4).

corresponding to the shift in datasets are reported in Table 8. Attributed to the non-robustness of CNNs to shift, there is a decrease in the overall accuracy for class-incremental tasks. However, we observe the decrease in incremental accuracy to be less for Gaussian-based OWM than OWM, with an average decrease of about 2% less. This illustrates a steady response to translation in the dataset and highlights the shift-invariant tendency of the Gaussian-based network.

**Statistical Significance of the Improvements Achieved by $S^2C^2IL$:** We have conducted t-tests to evaluate the statistical significance of the performance improvement achieved by the proposed $S^2C^2IL$ algorithm. Figure 6 (a) reports the p-values obtained when tested for the performance difference between OWM+CL and $S^2C^2IL$ on split-CIFAR10 and split-SVHN datasets, as reported in Table 1. Figure 6 (b) reports the p-values obtained when tested for the performance difference of the proposed algorithm with and without PWS on the same datasets, as reported in Table 7. For a confidence level of 0.05, we have observed that all the p-values obtained for both experiments are statistically significant. The performance gain between OWM+CL and $S^2C^2IL$ when trained on split-CIFAR100 dataset for 2 and 5 incremental tasks is also significant (p-values of 0.0058 and 0.0056, respectively). However, the performance gain of the proposed algorithm when trained on split-CIFAR100 dataset with and without PWS is not statistically significant.

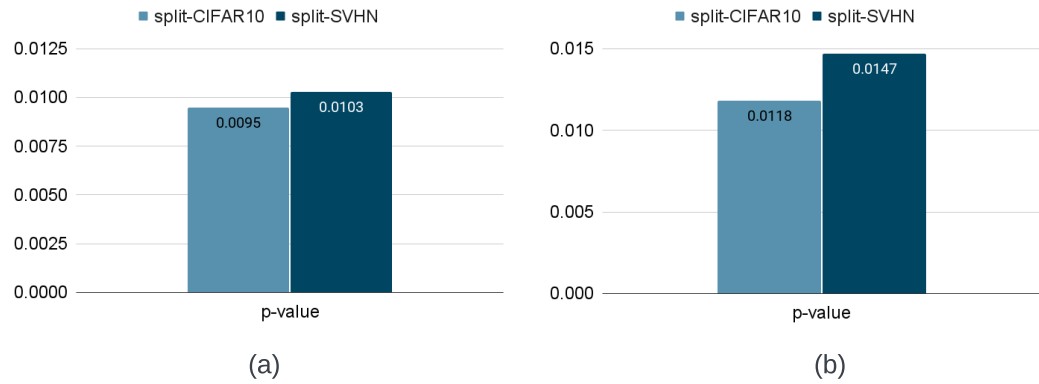

Figure 6: The p-values for t-test performed to test the statistical significance of the improvement in the performance of (a) OWM+CL and $S^2C^2IL$ for the split-SVHN and split-CIFAR10 dataset; (b) the proposed $S^2C^2IL$ algorithm when trained with and without PWS for the split-SVHN and split-CIFAR10 dataset. All are reported at confidence score of 0.05.

**Performance of S²C²IL on High-Resolution Datasets:** We test the proposed algorithm on high-resolution datasets, namely STL-10 (Coates et al., 2011) and ImageNet-100 (Russakovsky et al., 2015). STL-10 contains images of size $96 \times 96$ from 10 classes. We divide it into 5 incremental tasks, each with 2 classes. We call this split-STL10 dataset. On this dataset, our algorithm achieves an average accuracy of $49.26\% \pm 0.38\%$. ImageNet-100 is a subset of ImageNet with 100 classes with an image size of $224 \times 224$. We split it into 10 incremental tasks, each with 10 classes. On this dataset, our algorithm attains an average accuracy of $23.87\% \pm 0.31\%$. These results demonstrate the effectiveness of our algorithm on challenging high-resolution datasets as well.

# 6  Conclusion

In this research, we focus on the problem of regularization-based class-incremental learning. We address it through unsupervised pre-training and propose a novel pretext task that augments labels instead of the data. During downstream training, we transfer the convolution weights till the penultimate layers from the pre-training and design a smoothing-based curriculum. We find that through the incorporation of self-supervised learning and curriculum learning, we are able to improve the generalizability of the model in the continual learning paradigm. The augmentation of labels instead of data in the pretext task further improves the learning for the current task and decreases the resource requirements for training the model. The utilization of the smoothing-based curriculum further enhances the model's performance. The proposed S²C²IL algorithm with the Orthogonal Weight Modification (OWM) backbone achieves state-of-the-art results on four benchmark datasets: split-CIFAR-10, split-CIFAR-100, split-SVHN and split-TinyImageNet. The performance is also evaluated on two challenging high-resolution datasets: split-STL10 and ImageNet-100. The proposed algorithm can be appended with a memory component for future performance gains.

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
