# OpenReview forum: "$S^2C^2IL$: Self-Supervised Curriculum-based Class Incremental Learning"
_TMLR — Rejected by TMLR_

### Review · Reviewer_R1TU · 2023-05-26

**Summary Of Contributions:**

This paper studies the task of class incremental learning (CIL).  They propose a new self-supervised learning (SSL) approach to better regularize the model and aim to prevent catastrophic forgetting.

**Audience:**

Yes

**Claims And Evidence:**

No

**Requested Changes:**


Primary Concerns
* “Confused about the primary technical contribution SLA”
    * There is no description of exactly how the random labels are generated.  Because of this, I cannot further judge the validity of the proposed method.
    * Generally, in IL, the available data per task is small.  I’m not convinced that SSL in general is the right paradigm here.  The results in Table 3 does not show a big difference across 3 SSL method.  What about not using SSL at all?
* Lack of ImageNet experiments.
    * Recent papers seem to include ImageNet experiments (ref: https://paperswithcode.com/task/incremental-learning).
    * Experiments with higher-res images seem important to validate the proposed method.

Minor Comments on Presentation
* Citation style is wrong. Instead of “Mai et al (2022)”, it should be “(Mai et al., 2022)”.  Be mindful of `citet` vs `citep`.
* Discussion of existing work is not accompanied with citations.
* The information conveyed in the figures generally can be fitted into a much more concise figure.


**Strengths And Weaknesses:**

CIL is an important topic of research, and a better understanding of catastrophic forgetting is needed.

---

> ### Author Response · Authors · 2023-07-02
> **Detailed response to comments of Reviewer R1TU**
>
> We thank the reviewer for their feedback. Our response to the reviewer’s concerns are addressed below:
>
> ### Primary Concerns:
>
> 1. Regarding different steps in SLA
>
> >**Description of generation of stochastic labels:** In this research, we introduce Stochastic Label Augmentation (SLA), which focuses on augmenting makeshift labels rather than images. These labels are sampled from a uniform distribution for multiple tasks, with each task comprising several classes. The training process is conducted using a multi-task approach, where distinct fully-connected layers are utilized for each of the tasks
>
> (The explanation is updated on page 5, section 3.1).
>
> >**Motivation of using SSL in Continual Setting:** Unlike humans, continual learning models struggle to extract multiple important features from images. Their performance is optimized based on the classification objective function, resulting in the extraction of only the features necessary for the current task. This can cause a loss of prior information, as the model is unaware of the features required for joint classification with previously learned tasks. For instance, if a model learns to classify dogs and birds by counting their legs, it may lack sufficient information to classify a cat in a subsequent task. Without replaying previous inputs, the model cannot extract additional information about previous classes as they are no longer available. As more tasks are added in a continual learning setting, the model selects only the features necessary for current and subsequent incremental tasks, potentially leading to missing parts that affect joint classification accuracy. We term this phenomenon as Allocentric Ignorance. By using SLA-based pre-training, we enable the model to extract more meaningful features from the same set of images while preserving prior information and reducing training time and computational resources.
>
> (The explanation is updated on page 2, section 1, and page 5, section 3.1).
>
> >**Significance of Results in Table 3:** In Table 3 of the main paper, we observe an improvement of almost 4%-5% for the split-CIFAR10 dataset. To highlight that the improvement in the split-CIFAR10 dataset is significant, we perform t-tests on the performance obtained from the different pretext tasks and report the corresponding p-values below:
>
> >t(Rotation, SLA / split-CIFAR10) = 0.0006\
> >t(Colorization, SLA / split-CIFAR10) = 0.0013\
> >t(Rotation, SLA / split-SVHN) = 0.0171\
> >t(Colorization, SLA / split-SVHN) = 0.0465\
>
> >The statistical test shows that the difference in the performance of different pretext tasks at a confidence interval of 0.05 is statistically significant. As suggested by the reviewer, we have updated Table 3 to include the performance obtained without Self-Supervised pretraining. It is pertinent to note that without SSL, the performance is lowest on both datasets.
>
> (These observations are updated on page 13, section 5).
>
>
> 2. Experiments on ImageNet dataset.
> >**Results on ImageNet dataset:** Due to constraints in time and computational resources, we were able to conduct our experiments on the ImageNet-100 dataset with 10 incremental tasks. After running the experiments with 5 seeds, our proposed algorithm achieved a performance of 23.87% $\pm$ 0.31%. \
> >**Results on High-Resolution dataset:** The reviewer kindly suggested evaluating our algorithm on a high-resolution dataset as well. In response, we tested our algorithm on the split-STL10 dataset for 5 incremental tasks and achieved a performance of 49.26% $\pm$ 0.38% after testing our model with 5 seeds.
>
> (These experiments are incorporated in the main draft on page 16, section 5).
>
>
> ### Minor Edits
>
> 1.We thank the reviewer for pointing it out. We have corrected the format of citations in the updated draft (mostly on page 3 and 4, section 2 and minor edits in other part of the paper).
>
> 2. We have updated existing work with their corresponding citations in the revised draft (updated on pages 3 and 4, section 2).
>
> 3. As suggested by the reviewer, we have updated the draft with a concise figure on page 2.

---

### Review · Reviewer_hCXH · 2023-05-28

**Summary Of Contributions:**

The paper proposes an approach for class-incremental learning from the regularization-based continual learning perspective. The proposed system contains four components: (1) pre-training with the novel "stochastic label augmentation," which augments the input images with multi-task stochastic class labels, (2) Orthogonal Weight Modification to prevent catastrophic forgetting, (3) a Gaussian-noise-based curriculum, and (4) L2 regularization between the pre-training and fine-tuning parameters between tasks. The proposed approach is evaluated on four datasets (split-CIFAR10, split-CIFAR100, split-SVHN and splitTinyImageNet) and demonstrates competitive performance against the baseline approaches. The paper also performs ablation studies to confirm the effectiveness of Stochastic Label Augmentation, weight-sharing, and Gaussian noise.

**Audience:**

Yes

**Claims And Evidence:**

No

**Requested Changes:**

I encourage the authors to provide a comprehensive revision of the paper with a strong focus on why and how "stochastic label augmentation" works, in addition to improving the general writing style and technical clarity.

Please address my comments in the "Weaknesses" section. It is important because "stochastic label augmentation" is one of the paper's main contributions.

Writing style:

- The abstract should offer a high-level motivation for the proposed approach and the gap it aims to address in the context of the literature.
- The subsections in the methodology section are a bit isolated, making it difficult to understand how each component contributes in the context of class-incremental learning. For example, Sec. 3.2.1 describes OWM but unclear how it is used in the context of pre-training and fine-tuning within and between tasks. Sec. 3.3 is intended to be a high-level description of the approach, but a large portion of it contains details of a specific regularization technique.

Technical clarity:

- The authors should clarify whether standard deviations or confidence intervals are used across all tables.
- The paper lacks statistical rigour in that the best method is determined based on average performance and does not consider the variance across different methods. In Table 1, the difference between OWM + CL and S2C2IL does not appear to be significant for Split-CIFAR100. In Table 5, the results on split-CIFAR10 and split-CIFAR100 does not appear to be statistical significant, which undermines the argument on Penultimate Weight Sharing.
- I'm curious if there is any motivation for setting the regularization parameters a and b to 10 and 18—these are very large regularization parameters.
- Reproducibility: I encourage the authors to release the code early for better reproducibility.

**Strengths And Weaknesses:**

Strengths:
1. The proposed self-supervised learning approach, especially "Stochastic Label Augmentation," is interesting, relevant and novel.
2. The proposed approach demonstrates competitive performance against baseline methods, especially on the split-TinyImageNet dataset, where the model outperforms memory-based approaches by a large margin.
3. The paper presents interesting empirical findings on the impact of the penultimate layer and Gaussian noise on continual learning.

Weaknesses:
1. Motivation of "stochastic label augmentation": The paper should provide a convincing argument on why random label augmentation can serve as a good pre-training objective in the context of "garbage-in, garbage-out." The paper attempts to argue, “Pre-training the model with unlabeled data drives the model to extract more and more information from each input image as the model will be forced to minimize the loss for each of the tasks,” but overlooks the fact that the labels are random and do not carry semantic information of the input data.
2. Motivation of "multitask-learning" for "stochastic label augmentation": The paper proposes to use multitask-learning during the pre-training phase, but it is unclear how the task space is defined given the random nature of labels. It is difficult to understand how multitask learning would help as there is no clear inductive bias across tasks defined by random labels.
3. Technical clarity and writing style: The paper is difficult to read. The writing should be improved for better technical clarity (Please refer to more details in the "Requested Changes" section).

---

> ### Author Response · Authors · 2023-07-02
> **Detailed Response to comments of Reviewer hCXH (Part 1: Addressing Weaknesses)**
>
> We thank the reviewer for appreciating our work and providing feedback. Our responses to the reviewer’s concerns are provided below:
>
> ### Weaknesses
>
> 1.1. **SLA as a good pre-training objective:**
> >We agree with the reviewer that the proposed SLA task uses stochastically sampled labels, which are not semantically interpretable for humans per se. The pioneering work of [1] established that the models are capable of capturing meaningful information out of random labels even when there is no correlation between images and labels. Further research [2] has shown that, for a given pretext task, the initial layers of a model learn generalized semantic features, while the final layers learn task-specific features. Recent studies on training models with randomly labeled data ([3]) have revealed that the generalized information learned leads to a positive transfer of features, while specialization in later layers results in a negative transfer from pretext to downstream tasks.\
>  Inspired by the existing literature, we proposed an SLA-based pretext task in which the initial layers of the model extract generalized information and the final layer learns specialized information. To facilitate the positive transfer of knowledge from pretext to downstream tasks while preventing negative transfer, we employ Penultimate Weight Sharing (PWS), which skips the transfer of weights from the final section of the model. In summary, our approach aims to extract as much information as possible from each image [1] and transfer only the generalized information to downstream tasks, while excluding any task-specific information that may have been learned in the process.
>
> We have included the above explanation in the updated draft on Pages 12 and 13, section 5.
>
> 1.2.**The Motivation of "stochastic label augmentation:**
> >Continual learning models face the challenge of being unable to extract multiple significant features from images, unlike humans. Their performance is optimized based on the classification objective function, resulting in the extraction of only the features necessary for the current task. This can cause a loss of prior information, as the model is unaware of the features required for joint classification with previously learned tasks. For instance, if a model learns to classify dogs and birds by counting their legs, it may lack sufficient information to classify a cat in a subsequent task. Without replaying previous inputs, the model cannot extract additional information about previous classes as they are no longer available. As more tasks are added in a continual setting, the model becomes egocentric with the current data at hand and selects only the features necessary for current and subsequent incremental tasks, potentially leading to missing parts that affect joint classification accuracy. We term this phenomenon as Allocentric Ignorance. By using SLA-based pretraining, we enable the model to extract more meaningful features from the same set of images while preserving prior information and reducing training time and resource requirements.
>
>
> We have included the above explanation in the updated draft on Page 2 section 1 and page 5, section 3.1, for improved clarity of the readers.
>
> 2. **Motivation of "multitask-learning" for "stochastic label augmentation":**
> >The study from the authors of [1] demonstrates that models are capable of extracting meaningful information from images, even when presented with random labels. Building on this finding, we extended the SLA to a multi-task regime by augmenting each image with additional labels, thereby encouraging the model to extract even more information. As suggested by the literature [4], a multi-task learning setting can be more effective than a single-task setting, as it provides more supervision for each image and allows the model to learn from different tasks. Our empirical results support this conclusion, showing improved performance when scaling the model from single tasks to multiple tasks in pertaining.:
>
> | Number of Tasks (M)  | Performance (in %)  |
> | ------------- |:-------------:|
> | 1      | 59.93 ± 0.42     |
> | 2      | 60.82 ± 0.72     |
> | 3      | 61.64 ± 0.57     |
>
> Table 1: The demonstration of the effect of increasing the number of tasks (varying values of M) in the pretraining step of SLA on the performance of the proposed S2C2IL when trained on split-CIFAR10 for 5 incremental tasks.
>
> We have included the details of the above experiment in the updated draft on page 13, section 5.

---

> > ### Author Response · Authors · 2023-07-02
> > **Detailed Response to comments of Reviewer hCXH (Part 2: Addressing Requested Changes and Technical Clarity)**
> >
> > ### Addressing Requested Changes
> >
> > 1. We have modified the abstract as suggested by the reviewer in the updated version of the paper (updated on page 1 of the main paper).
> >
> > 2. The proposed algorithm of S2C2IL is broadly divided into two parts, first is the Stochastic Label Augmentation-based pretraining step, and the second is Curriculum learning-based downstream training step. As suggested by the reviewer, the methodology section in the revised paper has been organized as follows:
> > >**Section 3.1:** In this section, we discuss the problem of allocentric ignorance and propose a novel pretext task to alleviate it. The pretext task augments labels instead of images for learning feature-rich representations.\
> > >**Section 3.2:** In this section, we discuss the curriculum-based smoothing mechanism coupled with Orthogonal Weight Modification (OWM) backbone to mitigate catastrophic forgetting.\
> > >**Section 3.3:** In this section, we put everything together and discuss how the entire algorithm works in synergy to achieve state-of-the-art performance in a class incremental setting. We also discuss the proposed task-regularizer, which regularizes the weights of the model and keeps it within the norm bounds. This implicitly alleviates the problem of catastrophic forgetting.
> >
> >
> > ### Addressing Technical Clarity
> >
> > -> We report standard deviation of performance achieved over 5 random seeds across all the tables in the paper (Updated on page 11, Implementation Details of section 4).
> >
> > -> For comparisons, we followed the protocols mentioned in the existing literature. These research papers made use of average accuracy and standard deviation values, where each experiment was run for five random seeds.
> >
> > ->We have conducted *t-tests* to evaluate the statistical significance of the performance improvement achieved by the proposed S2C2IL algorithm. When tested for the performance difference between OWM+CL and S2C2IL on split-CIFAR10 and split-SVHN datasets, we obtained p-values of 0.0095 and 0.0103, respectively. When tested for the performance difference of the proposed algorithm with and without PWS on the same datasets, we obtained p-values of 0.0018 and 0.0147, respectively. For a confidence score of 0.05, we have observed that all the p-values obtained for both experiments are statistically significant. The performance gain between OWM+CL and S2C2IL when trained on split-CIFAR100 dataset for 2 and 5 incremental tasks is also significant (p-values of 0.0058 and 0.0056, respectively). However, the performance gain of the proposed algorithm when trained on the split-CIFAR100 dataset with and without PWS is not statistically significant.
> >
> > (We have added the above statistical tests on the pages 15 and 16, section 5 in the updated draft of the paper.)
> >
> > ->The regularization parameters a and b are hyperparameters and are empirically calculated values. These values are high because each incremental task brings in new data due to which the model weights quickly start to drift away from the previous task. Hence, to keep them bounded, the regularization constants are set high.
> >
> >
> > ->The code will be made publicly available upon acceptance and is currently available at: https://anonymous.4open.science/r/S2C2IL-Self-Supervised-Curriculum-based-Class-Incremental-Learning-8E81/
> >
> > References:\
> > [1]  C. Zhang, S. Bengio, M. Hardt, B. Recht, and O. Vinyals. “Understanding deep learning requires
> > rethinking generalization.” In International Conference on Learning Representations (ICLR), 2017.\
> > [2] I. Misra, and L. Maaten. "Self-supervised learning of pretext-invariant representations." Proceedings of the IEEE/CVF conference on Computer Vision and Pattern Recognition (CVPR), 2020.\
> > [3] H. Maennel, I. Alabdulmohsin, I. Tolstikhin, R. Baldock, O. Bousquet, S. Gelly, and D. Keysers, "What do neural networks learn when trained with random labels?." In Advances in Neural Information Processing Systems (NeurIPS), 2020.\
> > [4] A. Malhotra, M. Vatsa, and R. Singh. "Dropped Scheduled Task: Mitigating Negative Transfer in Multi-task Learning using Dynamic Task Dropping." In Transactions on Machine Learning Research (TMLR), 2022.

---

### Review · Reviewer_zDJZ · 2023-06-02

**Summary Of Contributions:**

This paper introduces S2C2IL to prevent catastrophic forgetting in continual learning. It employs a regularization-based approach with a novel pretext task using stochastically-augmented labels to enhance feature representations without relying on augmented image transforms. To prevent knowledge transfer from the pretext task, the final section of the pre-trained network is excluded during feature transfer. S2C2IL, trained with an orthogonal weight modification backbone, outperforms state-of-the-art regularization-based and memory-based class incremental algorithms on well-known datasets.








**Audience:**

Yes

**Broader Impact Concerns:**

No specific concern.

**Claims And Evidence:**

Yes

**Requested Changes:**

(1) The novelty of the proposed method is not very significant. it is suggested that the authors should provide a more detailed explanation of their innovation as the current method appears to be a combination of existing approaches.

(2) It is recommended to carefully review the symbols and language expressions in the paper to ensure their completeness and accuracy.

(3) Please provide a comparison of the method's time complexity to validate its efficiency in terms of time.

**Strengths And Weaknesses:**

Pros:

(1) Class Incremental Learning is a very fundamental problem for vision domains, and preventing catastrophic forgetting from the self-supervised learning perspective is a very interesting topic.

(2) The paper is well-organized and easy to be understood.

(3) Extensive experiments analysis is informative for the readers.

Cons:

(1) The overall innovation of the paper is relatively low and, to some extent, incremental. The methods lack novelty as they are a combination of existing approaches.

(2) What is A in Equation 2 that doesn't seem to explain its meaning.

(3) What is the training time/cost comparison for your whole method and corresponding baseline model? It would be very helpful if the authors could provide a specific training time/cost compared with the baseline methods.

Overall, I have some concerns about the novelty of the paper. The paper only combines several classic algorithms, and in its current version, it does not meet the requirements for publication in my opinion.

---

> ### Author Response · Authors · 2023-07-02
> **Detailed response to comments of Reviewer zDJZ**
>
> We thank the reviewer for their feedback. Our response to the reviewer’s concerns is addressed below:
>
> ### Requested Changes:
>
> 1. We would like to highlight that,  given the advances in self-supervised learning and curriculum learning in recent years, the exploration of a novel combination of the existing OWM approach as a backbone is not a weakness but an essential innovation of this work. The other components of the pipeline work in synergy, aiding OWM in the mitigation of catastrophic forgetting enhances the beauty of the algorithm. Each component contributes individually towards the performance of the proposed S2C2IL algorithm, which are also validated by the ablation experiments available in Table 2 of the main paper. We discuss them below in brief:
> > **Unsupervised Pre-Training by Stochastic Label Augmentation:** We propose a novel pretext task for feature extraction to alleviate allocentric ignorance. In this pretext task, we augment the labels instead of images to learn feature-rich representations with limited scaling of training time and computational resources.\
> > **Self-Supervised CIL with Task Regularization:** The model is trained iteratively for the pretext and downstream tasks with the proposed task regularization term to prevent catastrophic forgetting.\
> > **Curriculum for Downstream Training:** For the downstream task, we employ a curriculum-based smoothing mechanism in combination with Orthogonal Weight Modification (OWM).\
> > **Penultimate Weight Sharing:** The weights obtained after training the model on the pretext task are transferred to the downstream model with the exception of weights from the last layer.
>
> 2. We thank the reviewer for pointing it out. Matrix $A_l$ consists of all trained input vectors spanning the input space where the previous task has already been learned as its columns, e.g. $A_l = [a1, …a_l−1]$. We have updated the explanation in equation 2 (on page 6, section 3.2.1) and verified other equations as well.
>
> 3. As suggested by the reviewer, Table 1 reports the time taken for each pretext task, and Table 2 reports the time taken to train the recent state-of-the-art algorithms on the split-CIFAR100 dataset for 5 incremental tasks with 64 batch size.
>
>
> | __Technique__  | __Time Taken (in seconds)__  |
> | ------------- |:-------------:|
> | Rotation Pretext      | 9s     |
> | Colorization      | 8s      |
> | SLA      | __2s__     |
>
> Table 1: Time taken (in seconds) for an epoch by different pretext tasks on split-CIFAR100 dataset for 5 incremental task.
>
> We also computed the training time of the existing algorithms (from Table 1 of the main paper) for an epoch on split-CIFAR100 dataset for 5 incremental tasks. We observed that the existing algorithms such as MUC [1] and SSRE [2] require about 8 seconds and IL2A [3] and FeTrIL  [4] require over 200 seconds to train for a single epoch. The proposed S2C2IL algorithm takes 6 seconds for each epoch making it computationally efficient when compared to most of the existing baseline algorithms.
>
>
> (This experiment is also added in the updated draft of the paper on page 14.)
>
> References:\
> [1] Liu, Yu, et al. "More classifiers, less forgetting: A generic multi-classifier paradigm for incremental learning." Computer Vision–ECCV 2020: 16th European Conference, Glasgow, UK, August 23–28, 2020, Proceedings, Part XXVI 16. Springer International Publishing, 2020.\
> [2] Zhu, Kai, et al. "Self-sustaining representation expansion for non-exemplar class-incremental learning." Proceedings of the IEEE/CVF Conference on Computer Vision and Pattern Recognition. 2022.\
> [3] Zhu, Fei, et al. "Class-incremental learning via dual augmentation." Advances in Neural Information Processing Systems 34 (2021): 14306-14318.\
> [4] Petit, Grégoire, et al. "Fetril: Feature translation for exemplar-free class-incremental learning." Proceedings of the IEEE/CVF Winter Conference on Applications of Computer Vision. 2023.

---

### Review · Reviewer_W3C7 · 2023-06-02

**Summary Of Contributions:**

This paper introduces a regularization-based CL algorithm that is composed of two steps per task, one self-supervised pretraining named as stochastic label augmentation) and then training for the downstream task in a CIL setting where they regularize the weights using a curriculum-based smoothing mechanism and orthogonal weight modification. The authors evaluate their method on four relatively small scale datasets using a very small 4-layered CNN model.



**Audience:**

Yes

**Claims And Evidence:**

No

**Requested Changes:**

Please read the weaknesses above.

**Strengths And Weaknesses:**

Strength:

1- The paper os well-written and easy to follow with clear diagrams and algorithms.
2- The idea of pretaining with the pretext tasks of augmented labels as opposed to data is interesting.

Weaknesses:
Overall I think the idea behind the paper could be interesting a few years ago and with all the advances in the field of CL approaches it seems a bit outdated.
1. My main concern is that I am not convinced why one should use a regularization based method in CL while there are parameter-efficient and memory-free or memory-inexpensive methods introduced such as [1-3]. Especially that the method is evaluated on too easy datasets such as CIFAR10/100 and SVHN using a 3-layer CNN. I am curious to know the motivation behind using such a small network. The results reported for baselines are way lower than their originally reported values due to this I think. Does that mean you tuned all the baselines for such a small network?

2. Another concern is regarding the pretext training phase. If each task needs pretext training, how is this method efficient?

3. Is task ID required when we test the model on a sample? Do we need to know M prior to starting the experiments?




[1] Wang, Z., Zhang, Z., Ebrahimi, S., Sun, R., Zhang, H., Lee, C. Y., ... & Pfister, T. (2022, November). Dualprompt: Complementary prompting for rehearsal-free continual learning. In Computer Vision–ECCV 2022: 17th European Conference, Tel Aviv, Israel, October 23–27, 2022, Proceedings, Part XXVI (pp. 631-648). Cham: Springer Nature Switzerland.
[2] Smith, J. S., Karlinsky, L., Gutta, V., Cascante-Bonilla, P., Kim, D., Arbelle, A., ... & Kira, Z. (2023). CODA-Prompt: COntinual Decomposed Attention-based Prompting for Rehearsal-Free Continual Learning. In Proceedings of the IEEE/CVF Conference on Computer Vision and Pattern Recognition (pp. 11909-11919).
[[3] Wang, Z., Zhang, Z., Lee, C. Y., Zhang, H., Sun, R., Ren, X., ... & Pfister, T. (2022). Learning to prompt for continual learning. In Proceedings of the IEEE/CVF Conference on Computer Vision and Pattern Recognition (pp. 139-149).

---

> ### Author Response · Authors · 2023-07-02
> **Detailed response to comments of Reviewer W3C7**
>
> We thank the reviewer for their feedback. Our response to the reviewer’s concerns are addressed below:
>
> 1. The first point is divided into multiple parts and is addressed below:
> > 1.1. **Relevance of regularization-based methods over prompt-based methods:** The references shared by the reviewer point towards the Prompt-based algorithms, which are a relatively new set of algorithms in the arena of  continual learning. We agree with the reviewer that these are memory-inexpensive algorithms; however, either they are coupled with a small memory from previous tasks or at least carry a small memory of prompts to generalize on newer incremental tasks. Though they do pose an elegant new paradigm of continual learning, they still are not memory-free and need to store prompts beforehand, which is not the case with regularization-based algorithms.\
> > 1.2. **Motivation behind using small backbone network:** Regularization-based algorithms are still more practical to deploy in memory constrained settings as opposed to other methods. To deploy a network on edge devices like smartphones or surveillance cameras, the network is not allowed to have a lot of depth. Hence, following the protocols from [1] and [2] (references below), we have also adopted a 3-layer CNN as our backbone network.\
> > 1.3 **Datasets used for evaluation of the proposed algorithm:** To follow the protocols established by the existing methods and for a fair comparison with the existing state-of-the-art algorithms, we evaluated the proposed algorithm on split-CIFAR10, split-100 and split-SVHN datasets. To test our algorithm on a complex, large-scale dataset, we also report the performance on split-TinyImageNet dataset (in Table 4 of the main paper) and our algorithm outperformed the existing algorithms by a large margin. Since different papers used different backbone networks, the algorithms highlighted with * were re-run with the same backbone network for a fair comparison (mentioned in Table 1 of the main paper).\
> > 1.4 **Comparison with recent algorithms:** We have compared the performance of the proposed algorithm with several recent algorithms. Tables 1 and 4 in the manuscript highlight the comparisons across different datasets.
>
> 2. Traditional self-supervised algorithms augment data, leading to a significant increase in training time and computational resources. Though SLA also undergoes the cycles of pretraining and downstream training for each incremental task, we propose to augment labels instead of data, which incurs less training time and computational resources as opposed to other self-supervised algorithms. Experiments supporting this claim are included in section 5 of the revised manuscript at Page 14.
>
> 3. The task ID is not required while testing the model on any sample. M is a hyperparameter. In our experiments, we kept M as 3.
>
> References:\
> [1] G. Zeng, Y. Chen, B. Cui, and S. Yu. Continual learning of context-dependent processing in neural networks. In Nature Machine Intelligence, 2019.\
> [2] W. Hu, Z. Lin, B. Liu, C. Tao, Z. Tao, J. Ma, D. Zhao, and R. Yan. Overcoming catastrophic forgetting for continual learning via model adaptation. In International Conference on Learning Representations (ICLR), 2018.

---

### Decision · Action_Editors · 2023-07-31

**Recommendation:** Reject

**Comment:**

This is a promising paper and the authors made a good effort in the submission and in replying to reviewers. Unfortunately, the evidence presented in the paper to back up the main ideas was insufficient. I hope the feedback provided by 4 high quality reviews will help authors strengthen the paper in a submission to another venue.

**Audience:**

If the results were more convincing, it could be attractice for a large enough audience of TMLR.

**Claims And Evidence:**

Reviewers have reservations about performance of the proposed stochastic label augmentation (SLA), in particular because the results on ImageNet "seem to be substantially lower than existing literature".

More generally, all reviewers were not fully convinced about the paper claims, having also concerns about the scale of the datasets used. One of the reviewers questions the motivation for "multitask-learning" for "stochastic label augmentation", and another also feels that the paper has no good justification for why SLA works in practice.